# Validation of Lund University Sexual Harassment Inventory (LUSHI)—A Proposed Instrument for Assessing Sexual Harassment among University Employees and Students

**DOI:** 10.3390/ijerph192417085

**Published:** 2022-12-19

**Authors:** Per-Olof Östergren, Catarina Canivet, Gisela Priebe, Anette Agardh

**Affiliations:** 1Division of Social Medicine and Global Health, Department of Clinical Sciences Malmö, Lund University, 221 00 Lund, Sweden; 2Department of Social and Psychological Studies, Karlstad University, 651 88 Karlstad, Sweden

**Keywords:** sexual harassment, occupational health, student health, gender and health

## Abstract

The objective was to investigate the validity and reliability of a new instrument assessing sexual harassment at a public university in Sweden. In-depth interviews and focus group discussions resulted in a 10-item instrument, the ‘Lund University Sexual Harassment Inventory’ (LUSHI). A survey was sent to all staff, including PhD students, and students, with a response rate of 33% (*n* = 2736) and 32% (*n* = 9667), respectively. Exploratory factor analysis and Cronbach’s alpha statistics were applied. Having experienced one or more of 10 specific behaviors was defined as sexual harassment exposure and was reported by 17.1% of staff/PhD students and 21.1% of students. Exploratory factor analysis yielded two factors with Eigenvalues above 1, labeled ‘unwanted sexual attention of soliciting type’ and ‘unwanted sexual attention of non-soliciting type’. Rape/attempted rape fell outside of the two factors. The Cronbach’s alpha values of the original 10-item scale and of the two newly formed scales were 0.80, 0.80, and 0.66, respectively. The mentioned statistics were markedly similar among men, women, and non-binary individuals and between staff/PhD students and students. We conclude that the 10-item instrument could be used for assessing sexual harassment in university settings or any type of workplace.

## 1. Introduction

Sexual harassment (SH) is a well-established work-related risk for health, especially with regard to poor mental health [1,2,3]. Students in higher education institutions are also affected [4,5]. However, despite several decades of research concerning SH, there is still no real consensus regarding how to define the concept and thus no generally agreed upon instrument for measuring its prevalence [6,7]. This makes it very problematic to make comparisons across the many studies that have been performed, to reach a common understanding about the best ways to intervene against SH at a workplace, and, finally, to detect a potential change in the prevalence of SH at a follow-up after an intervention. 

The main difficulty regarding the assessment of SH seems to be the question of whether the instrument should be based on the legal definition of SH or on the psychological/emotional experience [8]. The legal definition within one and the same country or state is usually very clear, but it might change over time and varies between countries. This makes comparisons difficult both over time and between studies in different countries. Most investigators prefer to also include definitions of SH that capture the psychological experience, which seems necessary for making causal inferences regarding health, especially mental health. The inclusion of the psychological experience may well be warranted, since many SH situations are located in a wide ‘grey zone’ between what is clearly legally defined or fall outside of these boundaries. 

Moreover, there seems to be a considerable contextual influence on what is regarded as SH. As an illustration of this, an European Union-wide survey (using the same instrument for assessing SH in all participating countries) revealed a seemingly paradoxical pattern, namely that the prevalence of reported SH was considerably higher in European Union countries that score high on the gender equality index, compared to countries characterized by a low score [9]. It was suggested that this counter-intuitive observation could be due to the fact that greater awareness of gender inequality and gender rights in a particular population results in a broader definition of behaviors that are deemed to be SH. Therefore, strong argumentation for a contextual influence on what is perceived as SH makes the definition of SH in a population a ‘moving target’, something which may increase the importance of using updated, clear, and validated measures in instruments tapping the prevalence of SH. 

Power emerging from gender relations could also be an explanation for the finding that sexual advances in the workplace may be perceived as threatening and thus as harassment by most women, while not by most men [8]. Feminist theory postulates that sexual harassment could be regarded as a part of a constant negotiation concerning definitions of femininities and masculinities [10]. This means that men’s sexual harassment of women tends to reinforce a subordinate and passive feminine role, which is incongruent with working women’s ‘efforts to view themselves, and be seen by others, as dignified and equal employees’ [11]. The same implication would not necessarily be true in, for instance, the situation of a woman making sexual advances to a man. On the contrary, men may state that they are positively stimulated by this behavior in the workplace, or, as mentioned by one participant in a study on SH of men, if these advances are seen as bothersome, he has no difficulty addressing the woman in question to make it stop [10]. In order to be able to evaluate whether a particular SH event has a similar impact regardless of gender, an ideal instrument tapping SH should be validated in a population of mixed genders.

The influence of contextual factors on the definition of sexual harassment thus implies a continuous arbitration regarding where the limit should be drawn. This makes it important not only to register the experience of certain behaviors/situations in questionnaires and interviews, but to clearly include information about whether they are regarded as unwanted or not, as well as the gender of the victim and preferably also the perpetrator.

In conclusion, it seems very different to assess SH in a population or at a workplace in the same way as other important health risks, e.g., air pollution, noise or repetitive movements, where the biological response is given for a defined dose of a well-defined exposure and where comparisons of exposure over time or between populations are unproblematic.

However, despite the difficulties discussed above, there seems to be consensus in the scientific community that exposure to SH constitutes a significant risk for poor health that needs to be dealt with by increasing knowledge about its prevalence and the mechanisms underlying the negative effects on health. Therefore, the best available instruments should be used and continuously evaluated and developed in order that their virtues and fallacies may be discussed in relation to the interpretation of any empirical results.

One of the oldest and most widely used instruments for assessing SH at the workplace is the Sexual Experiences Questionnaire (SEQ), introduced by Fitzgerald et al. in 1988 [12]. This questionnaire, which contains a list of potentially offensive behaviors, has undergone a series of evaluations over time, resulting in a reduction in items from 50 to 17 in the most recent versions of the instrument. A short version of only eight items also exists [13].

In 1995, Fitzgerald et al. suggested that a ‘tripartite’ version of the SEQ, consisting of the three concepts *gender harassment, unwanted sexual attention,* and *sexual coercion*, achieved the best fit with regard to theory and statistic evaluation, while at the same time, constituted a conceptual link between the psychological and legal constructs of SH. *Gender harassment*, often expressed as comments, jokes, gestures, etc., has nothing to do with sexual desire or with a wish to ‘elicit sexual cooperation’ [14], but rather with traditional misogyny and with using women’s sexuality or alleged sexuality in inappropriate settings as a means to intimidate and degrade them [2,15,16]. The same kind of derogatory and hostile attitudes may be directed toward LGBTQ individuals and at men who deviate from the masculinity norm [17]. Together with *unwanted sexual attention, gender harassment* would reflect the (American) legal concept of ‘hostile work environment’. *Sexual coercion* (bribing or issuing threats of negative consequences if sexual invitations are not accepted) would correspond to the legal term ’quid pro quo’ [14]. 

The SEQ has over time been used in a number of workplace settings, often with minor or major adaptations [5], for instance, in order to fit specific situations, such as for military staff [18] or medical students [19]. A version specifically for male respondents was developed in 1996 [10] and another for Latina women in 2001 [20]. 

In 2004, Gutek et al. presented a thorough criticism of the SEQ, claiming that the many numerous versions precluded any entitlement of being a standardized instrument, that the subscale *sexual coercion* showed questionable evidence of reliability, and that all subscales were highly correlated with each other, which could imply that they are not sufficiently distinguishable from each other [13]. More importantly, the psychological aspect of the described offensive behaviors as being ‘offensive, unwelcome, and unreciprocated’ for the recipient [14] was not in fact consistently indicated in the questionnaire. Furthermore, for several of the items, it was ‘clear … (*that they*) … do not describe sexual behavior’, as, for instance, in ‘said things to put women down, for example that women don’t make good supervisors’ [13]. Finally, according to Gutek et al., the ‘criterion’ item—‘I have been sexually harassed’—is affirmed by considerably fewer than those who will affirm at least one of the SEQ items, supposedly demonstrating that the use of SEQ will overestimate the prevalence of the actual SH [13].

Regarding the latter objection, it is well documented that any inquiry detailing descriptions of potentially offensive behaviors will yield a higher prevalence than self-labeling of SH with a single item. This might be due to an unwillingness to admit victimization [7] or to the normalization of harassment behavior in the work environment [21,22]. 

Despite this criticism, the SEQ or, rather, the theoretical framework built by Fitzgerald et al., is repeatedly cited as fundamental [21]. Therefore, it has inspired the development of other questionnaires, such as the Bergen Sexual Harassment Scale [7] and the questionnaire used in a recent Norwegian country-wide study on SH among university students [23]. 

However, in both of these questionnaires, the concept of *gender harassment* is completely lacking. As noted by Timmerman and Bajema [6], this is more often the case in research originating outside of the United States. According to Swedish law, there is a distinction between ‘harassment’ in general and ‘sexual harassment’ [24]. Therefore, this conceptualization of SH may be more consistent with the perceptions of the general public regarding SH—even if these perceptions also differ by gender and by age [17,25].

A reasonable conclusion of the description of the state of the art seems to be that a true ‘gold standard’ for assessing SH does not currently exist and might not even be possible to develop in the future, due to the contextual dependency of how SH is perceived. Furthermore, SH is a controversial and significantly ideologically laden subject, both in a general context and from a scientific point of view. However, a commonly occurring nihilism, characterized by emphasis on the fact that very few studies have used an instrument yielding comparative results, may seriously hamper the growth of knowledge in the area of workplace SH. A constructive way out of this for new studies on SH would be to perform a critical validation of the instrument used, that addresses as many as possible of the issues raised above. 

’Tellus’, a project aimed at reinforcing preventive work against sexual harassment at Lund University, Sweden, was initiated in 2018. Individual interviews and focus group discussions were held with members of the target population [26], the results of which contributed to the development of a questionnaire for use in the subsequent university-wide survey targeting employees and students at all levels. We agreed on the following prerequisites for the questionnaire: It should (a) cover both ‘everyday’ SH and sexual assault, (b) cover both ‘traditional’ forms of SH and ‘new’ forms, such as online harassment, (c) clearly indicate that the behavior was ‘unwanted’, and (d) be inclusive of the experiences of, and possible to use, for men and LGBTQ individuals. 

Regarding the selection of the items of the instrument, on the one hand, we chose to align with previous research outside of the United States and, as described above, with the definition of SH in Swedish law, which, in turn, corresponds to public opinion, by restricting the behaviors indicated to those with a clear sexual connotation. This was clearly specified in the instructions to the respondents to the survey. On the other hand, we used the expression ‘unwanted comments’ as one of the items. This would allow respondents to include various offensive remarks, which for them had a sexual connotation. For the choice of items describing other specific behaviors, we were inspired by the questionnaire used by Phillips et al. [27]. The aim of this study is to describe the instrument and to address issues of construct validity. 

## 2. Methods 

The ‘Tellus’ project at Lund University, Sweden, was initiated in 2018. Two survey questionnaires were shaped, one for university staff (including PhD students who have formal employment in the Swedish system) and another one for students. After permission from the Vice chancellor’s office, e-mail addresses for all university staff and students were obtained from the university administration. The original items in English were translated by the research team in a process involving both native Swedish and native English speakers. Before sending out the questionnaire, face validity and feasibility of the SH items were discussed against the background of interviews and focus group discussions, which had been made by members of the core research team as another part of the Tellus project, as well as against comments from a small pilot sample of employees and students. This resulted in some very marginal linguistic changes.

The survey forms were sent out in November 2019. The response rate was 33% for university staff and PhD students and 32% for students. After exclusion of those with missing data on both sex and gender (see below), age (N = 9 and 46, staff/PhD students, students, respectively), and those who did not answer any of the 10 questions on experiences of sexual harassment (N = 4 and 74, staff/PhD students, students, respectively), the final study population consisted of 2736 university staff and PhD students and 9667 students. 

Two questions concerning the gender of the respondent were asked in the survey: ‘What gender were you assigned at birth?’ (female/male) and ‘what is your current gender identity?’ (female/male/I do not identify as male or female). We used the answers to the second question to categorize participants as woman, man, or non-binary gender; however, in cases with a missing answer on this question (N = 15 for staff/PhD students, N = 84 for students), the answer to the first one was applied. Those who had refrained from answering both questions were excluded from the analyses (N = 3 for staff/PhD students, and N = 69 for students).

Age was categorized into groups, separately for staff/PhD students and students. Country of birth was recorded as ‘Sweden’, ‘Nordic countries (outside Sweden)’, ‘Europe (outside Nordic countries)’, or ‘outside Europe’. Professional group was specified according to nine types, which were then aggregated into five categories, ‘professors‘, ‘senior lecturers’, ‘lecturers and researchers’, ‘PhD students’, and ‘administrative and technical support staff and others’. Students self-reported as ‘international student’, yes or no. 

The following text introduced the survey section about experiences of sexual harassment (SH). ‘We will now ask some questions about your experiences of sexual harassment and sexual violence. Sexual harassment is defined as conduct of a sexual nature that violates someone’s dignity. This can be, for example, through comments or words, groping or indiscreet looks. It can also include unwelcome compliments, invitations, or suggestive acts. Sexual violence is defined in this study as attempts to conduct, or the conduct of sexual acts in which the person did not participate voluntarily. Have you experienced any of the following situations during your employment/your time as a student at Lund University?’ 

This text was followed by descriptions of 10 different behaviors, with the answer alternatives in each case: ‘Yes, once’, ‘Yes, more than once’, and ‘No’. As mentioned in the Introduction, the choice of items was based on the results of the performed interviews and focus group discussions. Moreover, it was inspired by the list of items produced by Phillips et al.; however, the term ‘unwelcome’ was preferred before ‘inappropriate’ [27]. The following 10 items were included in the current SH scale: Unwelcome suggestive looks or gestures; unwelcome soliciting or pressuring for ‘dates’; unwelcome ’inadvertent’ brushing or touching; unwelcome bodily contact, such as grabbing or fondling; unwelcome gifts; unwelcome comments; unwelcome contact by post or telephone; unwelcome contact online, for example, social media or email; stalking; and ‘attempts to conduct or the conduct of oral, vaginal, or anal sex or other equivalent sexual activity in which you did not participate voluntarily’ (hereafter labeled ‘attempted or completed rape’). Those answering ‘Yes’ to at least one of these 10 questions were classified as exposed to experiences of SH and all others as non-exposed.

Spearman correlation was used to obtain a correlation matrix involving all 10 items. Exploratory factor analysis, using both oblique and orthogonal (Varimax) rotation, was performed in order to examine the relationships and to potentially discriminate between different aspects of sexual harassment. Factors with Eigenvalues of >1 were included. Thereafter, the reliability of the 10-item scale and of two sub-scales, derived from two of the factors, was assessed with corrected item—total correlations and Cronbach’s alpha. The correlation between the two new subscales was determined. Statistical significance was accepted at the level of *p* < 0.05. All analyses were performed using the IBM SPSS package, version 25. 

## 3. Results

The sociodemographic characteristics of the participants are presented in Table 1. The proportion of staff/PhD students who affirmed ever having experienced at least one of the mentioned behaviors at Lund University, and thus by our definition were exposed to sexual harassment (SH), was 17.1%. The corresponding figure for students was 21.1%.

Table 2 shows the frequencies of having experienced each of the 10 behaviors, by group and by gender. The three most common behaviors reported among staff/PhD students were, in descending order, unwelcome comments, unwelcome suggestive looks or gestures, and unwelcome ‘inadvertent’ brushing or touching. The pattern concerning the type of behavior reported was similar for women and men, but the prevalence was significantly higher among women. 

Among female students, the same behaviors were reported, but in a different order. Thus, unwelcome suggestive looks or gestures was the behavior most frequently affirmed. The most frequently reported behavior among male students was instead unwelcome bodily contact, such as grabbing or fondling, while both unwelcome suggestive looks or gestures and unwelcome ‘inadvertent’ brushing or touching were affirmed by 4.7%. Persons defining themselves as non-binary reported the same three behaviors as staff/PhD students, and in the same order, although with higher prevalence.

Table 3 shows that all the items of the scale were significantly correlated with each other. A pattern was observed when looking at a high level of correlation, i.e., Spearman coefficient of 0.3 or more, implying that certain clusters of items were more highly correlated than could be expected by chance, with the exception of the ‘attempted or completed rape’ and stalking items which did not correlate highly with any of the other items of the scale. This pattern warranted the next step in the analysis, i.e., a formal factor analysis.

Table 4 shows the exploratory factor analysis, stratified by staff/PhD students and students. Two factors consistently emerged, which in the non-stratified sample explained 50.0% of the variance. We considered four items that loaded on factor 1 to be representative of unwanted sexual attention behavior of a ‘general’ type, while five items that loaded on factor 2 seemed to imply a more personally targeted, or ‘soliciting’, unwanted sexual attention. 

One item loaded with approximately the same level of pattern coefficient value (0.46–0.53) on both factors. This item, ‘unwelcome soliciting or pressuring for ‘dates’, was deemed on face value to relate to factor 2, which we thus labeled as ‘unwanted sexual attention of soliciting type’, and factor 1, consequently, was labeled as ‘unwanted sexual attention of non-soliciting type’.

A third factor emerged within the group of staff/PhD students only. Due to this fact and to the relative infrequency of the two corresponding items, stalking and attempted and completed rape (1.4 and 1.3%, respectively), this factor was not further considered.

The reliability of the original 10-item scale and of the two newly formed scales, non- soliciting SH and soliciting SH, is presented in Table 5. The Cronbach’s alpha values were 0.80, 0.80, and 0.66, respectively. There were no items that would have increased the Cronbach’s alpha values, had they been deleted. The analyses were repeated, stratified by gender, with the following Cronbach’s alpha for the entire scale: 0.80 for women, 0.76 for men, and 0.86 for non-binary persons; for the non-soliciting scale: 0.81 for women, 0.71 for men, and 0.81 for non-binary persons; and for the soliciting scale: 0.65 for women, 0.71 for men, and 0.72 for non-binary persons. As for the ‘corrected item—total item correlation’ values, these were well above 0.5 for all the items of the non-soliciting scale, while slightly lower for the items of the soliciting scale. The latter items were also associated with slightly lower values for the 10-item scale (except for ‘unwelcome pressure about meeting/date’; value of 0.57). 

The correlation coefficients between the two new subscales were 0.37 for the entire study group, out of which, for women 0.37, for men 0.36, and for non-binary persons 0.40; for staff/PhD students: Women 0.35 and men 0.26; and for students: Women 0.35 and men 0.38.

The population distribution of having experienced the two types of SH was also analyzed (data not shown). Experiencing ‘non-soliciting SH’ was nearly two to three times more common than ‘soliciting SH’, among staff/PhD students and students, as well as across all three gender groups. 

## 4. Discussion

The results of this study showed that the 10-item scale used for measuring sexual harassment among university staff and PhD students and students, i.e., the Lund University Sexual Harassment Instrument (LUSHI), had good reliability both among staff and students, regardless of gender. However, when performing a formal factor analysis, two factors with Eigenvalues above one emerged, both among staff and students. The first factor comprised four of the scale items and the second factor five items. Considering the current discussion concerning the SH phenomenon, they were labeled ‘unwanted sexual attention of non-soliciting type’ and ‘unwanted sexual attention of soliciting type’. It was noted that there was a considerable overlap regarding exposure to the two factors. 

One item, assessing whether the respondent had experienced rape or attempted rape, was significantly correlated with all the other items of the scale, but fell outside of the factors produced in the factor analysis. While rape may be considered the ultimate form of ‘sexual harassment’, this more unusual and extreme event might more appropriately be classified as ‘sexual violence’. These two phenomena are explicitly identified as discrete, for instance in the review by Fedina et al. on campus sexual assault [28]. 

Our results could be compared with previous similar analyses of the SEQ or SEQ-like instruments reported earlier in the scientific literature. The early versions of SEQ consisted of five dimensions, which after analysis were reduced to three factors (*gender harassment, unwanted sexual attention,* and *sexual coercion*) [14]. However, as mentioned in the Introduction, in similarity to many other previous European studies of sexual harassment, we decided not to include items measuring general *gender harassment*. This aspect of SH was assessed in conjunction with another instrument in our university-wide survey, which tapped the prevalence of all types of discrimination, as defined by current Swedish law (i.e., discrimination based on age, gender, ethnicity, religion, sexual orientation, sexual identity, and disability). 

All items fell by and large within the SEQ-factor defined as ‘*unwanted sexual attention*’. The SEQ-factor ‘*social coercion*’, mainly defined by the legal notion ‘quid pro quo’ (sexual favors in exchange of something), was not represented by explicit items in our instrument, since this type of situation seemed rare and was not reported at all in the numerous interviews and focus group discussions that informed the discussions regarding our choice of survey instrument. However, the subscale ‘unwanted sexual attention of soliciting type’ may also tap dimensions of sexual pressure, i.e., the victim may perceive implicitly that accepting an individually directed sexual invitation or contact attempt could potentially give something in return (access to academic opportunities, funding, important network contacts, etc.), or similarly, i.e., that declining the invitation could mean being blocked from these resources.

Our 10-item scale covers two central aspects of sexual harassment and one important indicator of sexual violence. The first aspect, ‘non-soliciting SH’, could be regarded as an indicator of a more general climate of sexual harassment at a certain workplace, and different from ‘general’ gender harassment by its clearly sexual content. 

The other aspect, ‘soliciting SH’, indicates a more explicit effort to create a sexual relation between two individuals. This invitation may be experienced as particularly unwelcome if the initiative emerges from an individual who has a superior position in one of the prevailing power structures [8], (i.e., general gender-based power structures, other general power structures based on age, social class, ethnicity/race, etc.) or in formal or informal power structures specific to an academic workplace, or any combination of those. This issue will be further analyzed in forthcoming papers.

However, even if the psychometric properties of the soliciting subscale were deemed acceptable [29], they were somewhat weaker, with an overall Cronbach’s alpha value of 0.66 and with no items attaining the level of >0.5 for corrected item—total item correlation. It is possible that this subscale does not in fact represent a unified concept, i.e., with a common theoretical construct, but rather could be seen as a summative entity regarding this type of event; on ‘face value’, the items seem to belong together. This has implications regarding which items a user should choose to include in an instrument for assessing SH. In turn, this depends on the aim of a particular study, whether it seeks to determine the prevalence of SH or whether it seeks to capture the effect of specific aspects of SH. In the former case, it is important to consider that using many items for assessing the prevalence of SH in a population yields a higher and perhaps more true estimation of the prevalence of SH, since the items can trigger the memory in a more detailed manner than only an overarching question ‘have you experienced SH’ [21]. In the latter case, the slightly lower values of corrected item—total item correlation for two of the items could motivate their exclusion from the subscale assessing ‘soliciting SH’. Forthcoming studies of predictive validity, whether the scales in one form or another can predict important outcomes, should also be considered for decisions on which items should be included or not. 

We found it very interesting and potentially useful that the 10-item scale seemed to perform very similarly among staff/PhD students and students regarding reliability and construct validity, which implies that it could be used universally in different contexts, perhaps with minor adaptations. Furthermore, the types of SH reported among women, men, and non-binary persons were also strikingly similar, even if SH is more prevalent among women and non-binary persons. Nevertheless, the actual content of an SH experience, perhaps particularly in the case of ‘unwanted comments’, may still vary greatly according to the gender of both perpetrator and victim. Differences by gender may be even more pertinent when it comes to the impact of SH on the exposed person’s mental health. This aspect, predictive validity, was beyond the scope of this study but will be investigated in a forthcoming study.

One of the strengths of this study is that the chosen instrument was based on previous research concerning instruments used for assessing sexual harassment, as well as on a large number of individual interviews and focus group discussions that informed the decisions concerning the selection of the scale items. Moreover, the study covered all parts of the university organization, staff, as well as students. The fairly large sample allows for stability of the performed analyses even when stratified for gender groups. A participation rate of 32–33% could invite selection bias, but comparisons with register-based information concerning the composition of categories among staff/PhD students and students showed a high level of similarity, which should be a good indicator of the representativity of our sample.

## 5. Conclusions

The 10-item instrument (LUSHI) could be used as a tool for assessing sexual harassment in university settings or presumably any type of workplace, due to its demonstrated high reliability and acceptable construct validity. Furthermore, a formal factor analysis showed that the items could generate two subscales, ‘unwanted sexual attention of non-soliciting type’ and ‘unwanted sexual attention of soliciting type’. Future studies might show differences in the quantitative as well as qualitative impact of those constructs, e.g., on mental health, or that the impact differs according to the gender of the victim and/or perpetrator.

## Figures and Tables

**Table 1 ijerph-19-17085-t001:** Participants in the Tellus survey, Lund University, Sweden, 2020.

		Staff/ PhD Students		Students	
		Ns	%	Ns	%
Gender	Women	1551	56.7	6055	62.6
	Men	1161	42.4	3544	36.7
	Non-binary	24	0.9	68	0.7
	Total	2736	100.0	9667	100.0
Age groups (Staff/PhD students)	≤30	335	12.2		
	31–40	634	23.2		
	41–49	772	28.2		
	50–59	687	25.1		
	≥60	308	11.3		
Age groups (Students)	18–25			7488	77.5
	26–30			1280	13.2
	31–40			562	5.8
	≥41			337	3.5
Country of birth	Sweden	2115	77.3	7660	79.3
	Nordic country (outside Sweden)	100	3.7	229	2.4
	Europe (outside Nordic countries)	314	11.5	860	8.9
	Outside Europe	201	7.3	912	9.4
Professional group	Professors	286	10.5		
	Senior lecturers	385	14.1		
	Lecturers and researchers	475	17.4		
	PhD students	398	14.5		
	Administrative and technical support staff and Others	1190	43.5		
International student	Yes			1204	12.5
	No			8442	87.5
Experienced sexual harassment (SH) (based on 10-item scale)	Yes	469	17.1	2044	21.1
	No	2267	82.9	7623	78.9

**Table 2 ijerph-19-17085-t002:** Participants, presented as numbers and percentages, in the Tellus survey, who affirmed having experienced different kinds of sexual harassment, at least once since they arrived at Lund University.

	Staff/PhD Students				Students				All					
	Women (N = 1551)		Men (N = 1161)		Women (N = 6055)		Men (N = 3544)		Women (N = 7606)		Men (N = 4705)		Non-Binary (N = 92)	
	Ns	%	Ns	%	Ns	%	Ns	%	Ns	%	Ns	%	Ns	%
Unwelcome suggestive looks or gestures	231	14.9	28	2.4	1026	17.0	167	4.7	1257	16.6	195	4.2	20	21.7
Unwelcome soliciting or pressuring for ‘dates’	106	6.9	14	1.2	499	8.3	71	2.0	605	8.0	85	1.8	11	12.0
Unwelcome inadvertent brushing or touching	131	8.5	21	1.8	790	13.1	167	4.7	921	12.1	188	4.0	13	14.1
Unwelcome bodily contact, such as grabbing or fondling	80	5.2	17	1.5	665	11.0	191	5.4	745	9.8	208	4.4	8	8.7
Unwelcome gifts	38	2.5	8	0.7	66	1.1	12	0.3	104	1.4	20	0.4	1	1.1
Unwelcome comments	256	16.7	42	3.6	896	14.8	136	3.8	1152	15.2	178	3.8	20	22.2
Unwelcome contacts by post or telephone	52	3.4	18	1.6	130	2.2	23	0.7	182	2.4	41	0.9	5	5.4
Unwelcome contact online (e.g., social media, email)	60	3.9	18	1.6	400	6.6	55	1.6	460	6.1	73	1.6	6	6.5
Stalking	28	1.8	9	0.8	104	1.7	23	0.7	132	1.7	32	0.7	3	3.3
Attempted or completed rape	6	0.4	3	0.3	125	2.1	20	0.6	131	1.7	23	0.5	5	5.4
Experienced sexual harassment (SH) = at least one affirmative answer on the 10-item scale	380	24.5	81	7.0	1625	26.8	399	11.3	2005	26.4	480	10.2	28	30.4

**Table 3 ijerph-19-17085-t003:** Spearman’s correlations * between all 10 types of sexual harassment experiences. Tellus survey, N = 12,403.

	Looks or Gestures	Pressuring about ‘Dates’	‘Inadvertent’ Touching	Bodily Contact	Gifts	Comments	Contact by Post or Telephone	Contact Online	Stalking	Attempted or Completed Rape
Unwelcome suggestive looks or gestures	1.000	**0.439**	**0.533**	**0.440**	0.177	**0.598**	0.218	**0.330**	0.180	0.200
Unwelcome soliciting or pressuring for ‘dates’		1.000	**0.358**	**0.344**	0.274	**0.438**	**0.343**	**0.409**	0.246	0.218
Unwelcome inadvertent brushing or touching			1.00	**0.613**	0.152	**0.460**	0.201	0.280	0.157	0.237
Unwelcome bodily contact, such as grabbing or fondling				1.00	0.137	**0.372**	0.163	0.255	0.121	0.285
Unwelcome gifts					1.00	0.211	**0.309**	0.251	0.264	0.126
Unwelcome comments						1.00	0.242	**0.337**	0.215	0.194
Unwelcome contacts by post or telephone							1.00	**0.364**	0.297	0.166
Unwelcome contact online (e.g., social media, email)								1.00	0.288	0.202
Stalking									1.00	0.179
Attempted or completed rape										1.00

* All correlations are significant at the 0.01 level. Coefficients > 0.30 are shown in bold.

**Table 4 ijerph-19-17085-t004:** Exploratory factor analysis performed with Varimax rotation on the 10 SH items in the Tellus survey.

	Staff/PhD students			Students		All	
	Factor 1 ’Non-Soliciting’	Factor 2 ’Soliciting’	Factor 3 -	Factor 1 ’Non-Soliciting’	Factor 2 ’Soliciting’	Factor 1 ’Non-Soliciting’	Factor 2 ’Soliciting’
Unwelcome suggestive looks or gestures	**0.74**			**0.79**		**0.78**	
Unwelcome soliciting or pressuring for ‘dates’	0.51	**0.53**		**0.52**	0.46	**0.52**	0.47
Unwelcome inadvertent brushing or touching	**0.81**			**0.82**		**0.82**	
Unwelcome bodily contact, such as grabbing or fondling	**0.75**			**0.77**		**0.78**	
Unwelcome gifts		**0.67**			**0.67**		**0.66**
Unwelcome comments	**0.68**			**0.71**		**0.70**	
Unwelcome contacts by post or telephone		**0.72**			**0.69**		**0.71**
Unwelcome contact online (e.g., social media, email)		**0.63**			**0.60**		**0.59**
Stalking		**0.56**	0.53		**0.66**		**0.66**
Attempted or completed rape			**0.87**				
Initial Eigenvalue	3.7	1.3	1.1	3.7	1.4	3.7	1.3
Variance explained (%)	36.8	12.8	10.8	36.7	13.6	36.6	13.4

Factors with Eigenvalues > 1 and coefficients ≥ 0.40 are shown. If items load on several factors, the highest loading is shown in bold.

**Table 5 ijerph-19-17085-t005:** Reliability of the 10-item scale, the ‘non-soliciting sexual harassment (SH)’ subscale, and the ‘soliciting SH’ subscale. Tellus survey, N = 12,403.

	10-Item Scale			Factor 1 ’Non-Soliciting SH’			Factor 2 ’Soliciting SH’		
	Corrected Item—Total Correlation	Cronbach’s Alpha If Item Deleted	Overall Cronbach’s Alpha	Corrected Item—Total Correlation	Cronbach’s Alpha If Item Deleted	Overall Cronbach’s Alpha	Corrected Item—Total Correlation	Cronbach’s Alpha If Item Deleted	Overall Cronbach’s Alpha
Unwelcome suggestive looks or gestures	0.65	0.76	**0.80**	0.66	0.73	**0.80**			
Unwelcome inadvertent brushing or touching	0.62	0.76		0.66	0.73				
Unwelcome bodily contact, such as grabbing or fondling	0.55	0.77		0.57	0.77				
Unwelcome comments	0.62	0.76		0.59	0.77				
Unwelcome soliciting or pressuring for ‘dates’	0.57	0.77					0.48	0.60	**0.66**
Unwelcome gifts	0.30	0.80					0.38	0.64	
Unwelcome contacts by post or telephone	0.38	0.79					0.48	0.59	
Unwelcome contact online (e.g., social media, email)	0.47	0.78					0.50	0.57	
Stalking	0.31	0.80					0.38	0.63	
Attempted or completed rape	0.32	0.80							

## Data Availability

Data could be obtained from the corresponding author in a format which does not jeopardize the anonymity of the respondents.

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
