# Peer review of "Validation of Lund University Sexual Harassment Inventory (LUSHI)—A Proposed Instrument for Assessing Sexual Harassment among University Employees and Students"

_ijerph, 2022, doi:10.3390/ijerph192417085_

Round 1
Reviewer 1 Report
The manuscript title “Validation of Lund University Sexual Harassment Inventory (LUSHI) – a Proposed Instrument for Assessing Sexual Harassment among University Employees and Students” Overall, the subject matter of this paper is modern, interesting, and useful to the general reader. The authors wrote it concisely, to the point, and well supported by the literature. However, I consider this measurement tool to be a necessary and essential thing that is well needed in any department, university or general organization. Because of the work with a clear process to develop psychological measurement properties. about sexual harassment That can happen a lot in the organization. But may lack discussion or bring it to be an important issue of working in an organization that works normally. . I suggest some issue for author response below.
- I'm interested in explaining that, as a result, The Cronbach's alpha values of the original 10-item scale and of the two newly formed scales were 0.80, 0.80, and 0.66, for the last point identified 0.66 in this section. Or properties that are inferior to other 2 dimensions? Why? Please explain the limitations. or supporting reasons related to the behavioral measurement properties of this tool.
- The study sample was distributed. or is representative of education at the institutional level This university, however, I'm interested in applying it to an organization. or other agencies Should questions or issues be updated in more contextual detail? However, if the contextual, cultural difference, how will the use of this device be extended in practice to other educational institutions?
Overall, I am very satisfied with this excellent research. It can proudly spread to the academic society through this journal.
Author Response
Reviewer 1
The manuscript title “Validation of Lund University Sexual Harassment Inventory (LUSHI) – a Proposed Instrument for Assessing Sexual Harassment among University Employees and Students” Overall, the subject matter of this paper is modern, interesting, and useful to the general reader. The authors wrote it concisely, to the point, and well supported by the literature. However, I consider this measurement tool to be a necessary and essential thing that is well needed in any department, university or general organization. Because of the work with a clear process to develop psychological measurement properties. about sexual harassment That can happen a lot in the organization. But may lack discussion or bring it to be an important issue of working in an organization that works normally.. I suggest some issue for author response below.
- I'm interested in explaining that, as a result, The Cronbach's alpha values of the original 10-item scale and of the two newly formed scales were 0.80, 0.80, and 0.66, for the last point identified 0.66 in this section. Or properties that are inferior to other 2 dimensions? Why? Please explain the limitations. or supporting reasons related to the behavioral measurement properties of this tool.
We are grateful for the opportunity to clarify this. Scales could be of two types, those with items assessing an outcome having a common theoretical construct, e.g. measuring a construct like ‘depression’, versus those attempting to summarize an exposure to factors which lack a common theoretical construct, e.g. food items containing carbohydrates. In the first case one would expect a high degree of internal consistency, while in the latter case, a low degree. If an individual has a diet with a high intake of one particular type of food containing carbohydrates, it is likely that the intake of another type of food containing carbohydrates is lower, rather than higher. This will yield a low measure of internal consistency, but the measure with many such items would still be meaningful if the purpose is to measure the total intake of carbohydrates. In the text below, we have attempted to reason whether the SH scale items are of the first or the second type, which has implications for whether we should judge the value of the scales solely by the degree of internal consistency. Our conclusion is not to do so, and that forthcoming studies of the predictive value of the scales, e.g. regarding health outcomes will be important for a better assessment of the validity of the suggested scales. We have made an attempt to clarify this line of reasoning by a slight change of the wording in the following text in the Discussion section of the manuscript.
‘However, even if the psychometric properties of the soliciting subscale were deemed acceptable 29, they were somewhat weaker, with an overall Cronbach’s alpha value of 0.66 and with no items attaining the level of >0.5 for corrected item – total item correlation. It is possible that this subscale does not in fact represent a unified construct, i.e., with a common theoretical construct, but rather could be seen as a summative entity regarding this type of events; on ‘face value’, the items seem to belong together. This has implications regarding which items a user should choose to include in an instrument for assessing SH. In turn, this depends on the aim of a particular study, whether it seeks to determine prevalence of SH or whether it seeks to capture the effect of specific aspects of SH. In the former case, it is important to consider that using many items for assessing the prevalence of SH in a population yields a higher and perhaps more true estimation of the prevalence of SH, since the items can trigger the memory in a more detailed manner than just an overarching question ‘have you experienced SH’ 21. In the latter case, the slightly lower values of corrected item – total item correlation for two of the items could motivate their exclusion from the subscale assessing ‘soliciting SH’. Forthcoming studies of predictive validity, whether the scales in one form or another can predict important outcomes, should also be considered for decisions on which items should be included or not.’
- The study sample was distributed. or is representative of education at the institutional level This university, however, I'm interested in applying it to an organization. or other agencies Should questions or issues be updated in more contextual detail? However, if the contextual, cultural difference, how will the use of this device be extended in practice to other educational institutions?
The validation of the instrument has been performed using data from a large Swedish university with nine faculties, i.e. an unusually broad higher education institution, which is favorable for claiming reasonable validity across different types of higher education institutions. Concerning the wider contextual background, the more similar, i.e. other North-western European countries, the greater likelihood of generalizability. Since most items are judged to have high face validity, i.e. they describe very concrete situations, it is likely that the generalizability might be high across a reasonably wide contextual variation, albeit legislation might vary considerably.
Overall, I am very satisfied with this excellent research. It can proudly spread to the academic society through this journal.
We thank the reviewer for the very positive assessment of our work.
Reviewer 2 Report
Although sexual harassment is an important health and legal issue, there were several flaws in this study and limited its values.
1. “Sexual harassment (SH) is a well-established work-related risk for health, especially with regard to poor mental health.” It is obvious that sexual harassment occurs in the situations not limited to work-related. The authors described “sexual harassment in workplace” in Introduction for several times. I am not sure whether the instrument developed in this study will be applied in workplace only or in other situations also, especially that this study included students but not only employees.
2. “Students in higher education are also affected.” The meaning of the term “higher education” was ambiguous.
3. Giving example items of the instruments based on the legal definition of SH or on the psychological/emotional experience may help readers to understand.
4. The full-spellings for “EU” and “US” are needed.
5. Lines 56-67: “Moreover, there seems to be…arrive at an optimally clear definition.” In this paragraph the authors proposed that the levels of awareness of gender inequality and gender rights may result in a broader definition of behaviors that are deemed to be SH. However, its relationship with the necessity to develop a new instrument for assessing SH is not clear.
6. The same question existed in the next paragraph from lines 68-83.
7. I did not find how the authors resolve the problems of previous instruments caused by legal definition/psychological/emotional experience of SH, gender inequality and gender right in the processes of developing the new instrument.
8. How the authors adopted the 10 items from the focus group is not clear.
Author Response
Reviewer 2
Although sexual harassment is an important health and legal issue, there were several flaws in this study and limited its values.
- “Sexual harassment (SH) is a well-established work-related risk for health, especially with regard to poor mental health.” It is obvious that sexual harassment occurs in the situations not limited to work-related. The authors described “sexual harassment in workplace” in Introduction for several times. I am not sure whether the instrument developed in this study will be applied in workplace only or in other situations also, especially that this study included students but not only employees.
Two different questionnaires were used for employees and students, respectively, although the instrument for assessing sexual harassment was identical because of the need to be able to compare results between different settings. The psychometric properties of the instrument were investigated in a workplace academic context (we consider the university as the workplace for both staff and students). It could be used in other contexts than workplaces as well, but the items could be perceived differently, so we recommend a new investigation of reliability and validity for other contexts.
- “Students in higher education are also affected.” The meaning of the term “higher education” was ambiguous.
We are grateful for the opportunity to clarify this. We meant ‘higher education institutions’ – aka ‘tertiary level of education’, as defined by e.g., UNESCO.
Rephrased sentence in the first paragraph of the Introduction section
‘Sexual harassment (SH) is a well-established work-related risk for health, especially with regard to poor mental health 1-3. Students in higher education institutions are also affected 4, 5.
- Giving example items of the instruments based on the legal definition of SH or on the psychological/emotional experiencemay help readers to
Response The content of the instrument is based on discussions regarding a theoretical framework and scales used in previous research, which is a combination of the legal framework in place in most OECD-countries, viewed through the lens of 10 concrete experiences. The feasibility of those items was reviewed in a pilot study with employees and students at the same university as the major study and a few purely linguistic edits were made to increase the verbal understanding.
In the Swedish legal context SH is defined the following way.
‘Harassment may also be of a sexual nature. Besides comments and words, this could involve unwanted touching or leering. It could also be a question of unwelcome compliments, invitations or insinuations.’ See https://www.do.se/choose-language/english/what-is-discrimination#h-Harassmentandsexualharassment
In the questionnaire the following information, congruent with the legal definition, was given to the respondent.
‘We will now ask some questions about your experiences of sexual harassment and sexual violence. Sexual harassment is defined as conduct of a sexual nature that violates someone’s dignity. This can be, for example, through comments or words, groping or indiscreet looks. It can also include unwelcome compliments, invitations, or suggestive acts. Sexual violence is defined in this study as attempts to conduct, or the conduct of sexual acts in which the person did not participate voluntarily. Have you experienced any of the following situations during your employment/your time as a student at Lund University?’
The full quotation above is included in the Methods’ section of the manuscript.
- The full-spellings for “EU” and “US” are needed.
This has now been added in the manuscript.
Changed sentences in the Introduction section
‘Moreover, there seems to be a considerable contextual influence on what is regarded as SH. As an illustration of this, an European Union-wide survey (using the same instrument for assessing SH in all participating countries) revealed a seemingly paradoxical pattern, namely that the prevalence of reported SH was considerably higher in European Union countries that score high on the gender equality index, compared to countries characterized by a low such score 9.’
‘However, in both of these questionnaires, the concept of gender harassment is completely lacking. As noted by Timmerman & Bajema 6, this is more often the case in research originating outside the United States.’
‘Regarding the selection of the items of the instrument, on the one hand we chose to align with previous research made outside the United States and, as described above, with the definition of SH in Swedish law, which in turn corresponds to public opinion, by restricting the behaviors indicated to those with a clear sexual connotation.’
- Lines 56-67: “Moreover, there seems to be…arrive at an optimally clear definition.” In this paragraph the authors proposed that the levels of awareness of gender inequality and gender rights may result in a broader definition of behaviors that are deemed to be SH. However, its relationship with the necessity to develop a new instrument for assessing SH is not clear.
We thank the reviewer for the opportunity to clarify this.
Changed text in the Introduction section
‘Moreover, there seems to be a considerable contextual influence on what is regarded as SH. As an illustration of this, an European Union-wide survey (using the same instrument for assessing SH in all participating countries) revealed a seemingly paradoxical pattern, namely that the prevalence of reported SH was considerably higher in European Union countries that score high on the gender equality index, compared to countries characterized by a low such score 9. It was suggested that this counter-intuitive observation could be due to the fact that greater awareness of gender inequality and gender rights in a particular population results in a broader definition of behaviors that are deemed to be SH. Thus, strong argumentation for a contextual influence on what is perceived as SH makes the definition of SH in a population a ‘moving target’, something which may increase the importance of using updated, clear and validated measures in instruments tapping the prevalence of SH.
- The same question existed in the next paragraph from lines 68-83.
Changed text in the Introduction section
‘Power emerging from gender relations could also be an explanation for the finding that sexual advances in the workplace may be perceived as threatening and thus as harassment by most women, while not so by most men 8. Feminist theory postulates that sexual harassment could be regarded as a part of a constant negotiation concerning definitions of femininities and masculinities 10. This means that men’s sexual harassment of women tends to reinforce a subordinate and passive feminine role, which is incongruent with working women’s ‘efforts to view themselves, and be seen by others, as dignified and equal employees’ 11. The same implication would not necessarily be true in, for instance, the situation of a woman making sexual advances to a man. On the contrary, men may state that they are positively stimulated by such behavior in the workplace, or, as mentioned by one participant in a study on SH of men, if these advances are seen as bothersome, he has no difficulty addressing the woman in question to make it stop 10. In order to be able to evaluate whether a particular SH event has a similar impact regardless of gender, an ideal instrument tapping SH should be validated in a population of mixed genders.’
- I did not find how the authors resolve the problems of previous instruments caused by legal definition/psychological/emotional experience of SH, gender inequality and gender right in the processes of developing the new instrument.
We do not perceive this as a problem that easily could be resolved, the different perspectives on SH will always be present, since laws and emotions are different phenomena. However, it is important that the instruments are discussed against the legal background which is very important in a policy context, as well as from a perceptional-emotional point of view that mirror the actual experience of SH among individuals. We have looked over the sentences in the manuscript in which this is discussed and made some clarifications. We have tried to express this line of thinking in the following paragraph
‘However, despite the difficulties discussed above, there seems to be consensus in the scientific community that exposure to SH constitutes a significant risk for poor health that needs to be dealt with by increasing knowledge about its prevalence and the mechanisms underlying the negative effects on health. Therefore, the best available instruments should be used and continuously evaluated and developed so that their virtues and fallacies may be discussed in relation to the interpretation of any empirical results.’
- How the authors adopted the 10 items from the focus group is not clear.
We are also very grateful for the opportunity to clarify this.
Changed text in the beginning of the Methods section
The ‘Tellus’ project at Lund University, Sweden, was initiated in 2018. Two survey questionnaires were shaped, one for university staff (including PhD students who have formal employment in the Swedish system) and another one for students. After permission from the Vice chancellor’s office, e-mail addresses for all university staff and students were obtained from the university administration. The original items in English were translated by the research team in a process involving both native Swedish and native English speakers. Before sending out the questionnaire, face validity and feasibility of the SH items were discussed against the background of interviews and focus group discussions which had been made by the core research team as another part of the Tellus project, as well as against comments from a small pilot sample of employees and students. This resulted in some very marginal linguistic changes.’
Round 2
Reviewer 2 Report
The authors have revised their manuscript based on the reviewer's suggestions. I would like to suggest the editors accepting it for publication.